# Influence of ZnO Film Deposition Parameters on Piezoelectric Properties and Film-to-Substrate Adhesion on a GH4169 Superalloy Steel Substrate

**DOI:** 10.3390/mi13040639

**Published:** 2022-04-18

**Authors:** Guowei Mo, Yunxian Cui, Junwei Yin, Pengfei Gao

**Affiliations:** 1Mechanical and Electronic Engineering, College of Mechanical Engineering, Dalian Jiaotong University, Dalian 116024, China; moguowei00@163.com (G.M.); yjwzx@djtu.edu.cn (J.Y.); gaopengfei198935@163.com (P.G.); 2Department of Mechanical Engineering, Liaoning Machinery Electricity Vocational Technical College, Dandong 118000, China

**Keywords:** ZnO piezoelectric films, GSS substrates, deposition process parameters, crystal quality, magnetron sputtering

## Abstract

ZnO film is widely used in the field of health monitoring sensors, which has high requirements for the piezoelectric coefficient and film-to-substrate adhesion of the ZnO film. In this study, ZnO thin films were grown on a GH4169 superalloy steel (GSS) substrate using magnetron sputtering, and the effects of the sputtering power, argon–oxygen ratio, and sputtering pressure on the piezoelectric coefficient and film-to-substrate adhesion were studied. The composition, microstructure, and crystal orientation of ZnO thin films deposited under different process parameters were analyzed using X-ray diffraction (XRD), a scanning electron microscope (SEM), and an energy spectrum analyzer (EDS). The piezoelectric coefficient d_33_ was measured using a piezoelectric coefficient measuring instrument. The critical value of adhesion between the film and substrate was measured using the scratch method. The results demonstrated that the ZnO films had the most desirable properties when the sputtering power was 150 W, the argon–oxygen ratio was 25:10, and the sputtering pressure was 0.7 Pa. The XRD results showed that the ZnO film samples had the strongest (002) crystal orientation at 2θ = 34.4°; the SEM photos showed that the film samples were flat and uniform; and the EDS composition analysis results showed that the composition was close to the theoretical value. The maximum *d*_33_ coefficient value was 5.12 pC/N, and the maximum value of film-to-substrate adhesion between the ZnO films and GSS substrate was 4220 mN.

## 1. Introduction

ZnO is a semiconductor material with a wide band gap (3.37 eV), low dielectric constant, large electromechanical coupling coefficient, and excellent temperature stability [1,2,3]. As ZnO films have a highly preferred c-axis orientation, hexagonal wurtzite structure, and a resistivity that is generally larger than 10^−8^ Ω·cm, their piezoelectric properties have been widely studied and applied in many fields [4,5]. Various sensors have been developed according to the different properties of ZnO films, including pressure sensors [6], gas sensors [7], photoelectric sensors [8], temperature sensors [9], etc. At present, the main fabrication methods of ZnO films include magnetron sputtering [10], atomic layer deposition (ALD) [11], pulsed laser deposition (PLD) [12], molecular beam epitaxy (MBE) [13], the sol-gel method [14], chemical vapor deposition (CVD) [15], etc. Microstructure plays a significant role in the properties of ZnO thin films, which is why many researchers are absorbed in studying their crystal structure, surface morphology, and mechanical stability using different methods.

ZnO thin films had been fabricated by different approaches and applied in many fields. Ghosh et al. reported the influence of deposition time on the growth of ZnO nanopores at a fixed silicon (100) substrate temperature without the use of a catalyst [16]. Vakulov et al. investigated the stability of the resistivity of ZnO thin films deposited by PLD in a fixed temperature range [17]. It is very critical to study the compatibility of ZnO films, in addition to the effects of deposition time and deposition temperature. Hence, Ben Moussa et al. deposited ZnO sol-gel thin films by spin-coating onto Si samples at low temperatures and studied the effects of compatibility with CMOS and IC on the annealing process, surface topography, and piezoelectrical properties [18]. Based on the results of the research described above, the optimized CVD growth parameters of ZnO nanowires were investigated by Faisal, who synthesized single-crystal ZnO on an indium tin oxide (ITO)-coated glass substrate successfully and discussed the mechanism of the vertical growth model [19]. Similarly, Kumar et al. synthesized single-crystalline ZnO nanorods on a silicon substrate via a vapor–liquid–solid process using the CVD technique and optimized several parameters, such as substrate temperature, catalyst layer thickness, and reaction time [20]. However, in another study, ZnO piezoelectric thin films with high piezoelectric properties, uniform thickness, and low propagating loss were deposited using magnetron sputtering and were fabricated without multiple processes [21]. 

The deposition and characterization of ZnO thin films have been widely reported on different kinds of substrates. Molarius et al. investigated the effects of process parameters on the properties of ZnO films using RF magnetron sputtering and measured the piezoelectricity of ZnO films in the 1–2 GHz range on a Corning glass substrate [22]. Cimpoiasu et al. deposited ZnO films on a silicon wafer substrate using a DC sputter coater and discussed the aspects of stress in ZnO films and methods to prevent or reduce this stress [23]. Piezoelectric ZnO films have been widely applied in acoustic wave devices for the generation and detection of acoustic waves in non-piezoelectric substrates [24,25,26,27,28], such as SiO_2_, Si_3_N_4_, gold, aluminum, etc., and show excellent adhesion. In previously published work in the magnetron sputtering regime, various substrate materials, substrate temperature, deposition time, post-deposition annealing, and oxygen pressure have been studied. However, with the wide application of ZnO films in the field of health monitoring, it is important to study the process parameters of the deposition of ZnO films on GSS substrates. 

In the present work, the deposition of ZnO films on GSS substrates using DC pulsed magnetron sputtering is proposed. There are no public reports in the literature that describe sputtering piezoelectric ZnO films on GSS substrates. In order to better study the characteristics of ZnO films grown on GSS substrates, we carried out relevant studies. From the analysis of EDS, SEM, and XRD results of ZnO thin films, it was concluded that the element composition, surface morphology, and crystal orientation were greatly affected by the use of different deposition process parameters for magnetron sputtering. The influence of the sputtering power, argon–oxygen ratio, and sputtering pressure on the deposition ZnO thin films was investigated and discussed. Through the analysis of the piezoelectric coefficient d_33_ and film-to-substrate adhesion under different parameters, the best sputtering process parameters were obtained. These research results provide a theoretical basis for the fabrication of ZnO piezoelectric thin film sensors on a GSS substrate in addition to a technical reference for their application in the field of structural health monitoring.

## 2. Experimental Section

### 2.1. Deposition of ZnO Thin Films

ZnO thin films (1 μm) were deposited on two different substrates using the RF magnetron sputtering method: one (Figure 1a) was deposited directly on the GSS substrate, while the other (Figure 1b) was deposited out using a NiCr bottom electrode on a glass substrate—both were deposited using the same sputtering equipment (JZFZJ-500S) under the same processing conditions. The thickness of the NiCr electrode layer was 200 nm, and it was deposited using cathodic sputtering. The experimental materials and specifications are shown in Table 1. The properties of the GH4169 superalloy steel substrate are shown in Table 2.

The GSS substrate was polished to the mirror surface; then, both substrates were washed successively with acetone, absolute ethanol, and deionized water for 20 min and dried with nitrogen, and pre-sputtering was performed for 15 min followed by another 80 min of sputtering. The details of the sputtering parameters for the ZnO thin films are shown in Table 3.

### 2.2. ZnO Film Characterization

The factors affecting the fabrication of ZnO thin films by magnetron sputtering include the sputtering power, argon–oxygen ratio, sputtering pressure, and substrate temperature. In this experiment, we focused on the influence of the first three factors on the formation of films. ZnO film has a high piezoelectric coefficient and sufficient adhesion strength with GSS substrate, which is the premise for the film to be applied to health monitoring sensors. Therefore, the piezoelectric characteristics and film-to-substrate adhesion were selected as the major evaluation indexes of the experiment.

The crystallinity of the ZnO thin films was analyzed using an X-ray diffractometer (XRD, Empyrean sharp shadow). This experiment also analyzed the surface and cross-sectional morphology characteristics of ZnO films using scanning electron microscopy (SEM, JEM-2100F) and the composition of ZnO thin films with an energy dispersive spectrometer (EDS, GENESIS XM). An instrument developed by Foshan Zhuo Film Technology Co., Ltd. was used to measure the Piezoelectric coefficient d_33_; the measured value of the longitudinal piezoelectric coefficient was directly read out using special computer software, as shown in Figure 2. Figure 2a shows the principle of the quasi-static method. According to the piezoelectric effect, when the piezoelectric vibrator is subjected to an alternating external force that is far higher than its resonant frequency, it can produce alternating charge. In order to simplify the piezoelectric equation, it is assumed that the vibrator is not affected by the external electric field and only bears the force in the same direction as the polarization. The equation is as follows:(1)$$D_{3}=d_{33}T_{3}$$
(2)$$d_{33}=\frac{D_{3}}{T_{3}}=\frac{Q}{F}$$
where $D_{3}$ is the potential shift component, C/m^2^; $T_{3}$ is the longitudinal stress, N/m^2^; $d_{33}$ is the longitudinal piezoelectric strain constant, C/N or M/V; Q is the piezoelectric charge released by an oscillator, C; and F is the longitudinal low-frequency alternating force, N. As shown in Figure 2a, the charge released by the tested oscillator generates voltage on its parallel capacitors. At the same time, the charge released by the comparison oscillator generates voltage on its parallel capacitors. The following equation can be written from Formula (2):(3)$$\begin{array}{l} d_{33}^{\left(1\right)}=\frac{C_{1}V_{1}}{F}\\ d_{33}^{\left(2\right)}=\frac{C_{2}V_{2}}{F} \end{array}\}$$
where $C_{1}=C_{2}>100C^{T}$ (oscillator free capacitance).

Equation (3) can be further reduced to:(4)$$d_{33}^{\left(1\right)}=\frac{V_{1}}{V_{2}}d_{33}^{\left(2\right)}$$

In Equation (4), the value of the comparison vibrator $d_{33}^{\left(2\right)}$ is given, $V_{1}$ and $V_{2}$ can be measured, and the value of the measured vibrator $d_{33}^{\left(1\right)}$ can be obtained. If $V_{1}$ and $V_{2}$ are processed by an electronic circuit, the quasi-static value of the longitudinal piezoelectric strain constant $d_{33}$ of the measured vibrator can be obtained directly, as shown in Figure 2b.

The adhesion properties of the films were characterized using a scratch test. The instrument used for the scratch method was the Revetest scratch test system from the Swiss company CSM Instruments. The loading mode of the scratch instrument was continuous. The radius of the diamond needle was 200 μm, the loading speed was 4000 mN/min, and the load was gradually increased from 1000 mN to 5000 mN. The diamond indenter moves on the surface of the ZnO films. In this process, the vertical load is continuously increased until the film is destroyed. The load Lc corresponding to the destruction of the film is called the critical load—that is, the adhesion strength of the film. The failure behavior can be directly observed by microscope and tested using the acoustic signal method.

## 3. Results and Discussions

### 3.1. XRD Analysis

Figure 3 shows the XRD patterns of ZnO thin films deposited with different process conditions on the GSS substrate. The sputtering power, argon–oxygen ratio, and sputtering pressure of magnetron sputtering equipment are important process control parameters that affect the film-forming process of ZnO films, which directly determines the grain size and structure of ZnO films. The specific sputtering parameters are shown in Table 3. Figure 3a shows an XRD spectrum of ZnO films with different powers. Each sample corresponds to one sputtering power. The XRD results showed that all film samples had a (0002) crystal orientation at 2θ = 34.4°, which confirmed the development of a polycrystalline hexagonal-like structure. The structure promoted the growth of the ZnO films along the c-axis direction, which may have contributed to the improvement of their piezoelectric properties [29]. The full width at half maximum (FWHM) and grain size values are shown in Figure 3b; the grain sizes of ZnO film can be calculated using the Scherrer formula as follows:(5)$$D=\frac{k\lambda }{\beta \cos\theta }$$
where *D* is the grain size of ZnO (nm), *β* is the half height width FWHM (rad) of the ZnO (002) peak, *θ* is the Bragg diffraction angle, *k* represents the Scherrer constant (*k*= 0.89), and *λ* is the wavelength of the X-ray, i.e., Cu Ka radial (0.1541 nm). It can be observed that the FWHM was the smallest and the grain size (*D* = 10.7 nm) was the largest when the power was 150 W. Figure 3c,d clearly shows that when Ar:O_2_ = 25:10, there was only a (002) diffraction peak in the ZnO film samples. The peak, FWHM, and grain size *D* (*D* = 10.65 nm) represent the optimal state where the crystallization quality is best. Figure 3e,f shows that the best sputtering pressure was 0.7 Pa in the magnetron sputtering equipment; at this pressure, all indicators were relatively good. Hence, ZnO films may have desirable piezoelectric properties under these conditions (sputtering power: 150 W; Ar:O_2_ = 25:10; sputtering pressure: 0.7 Pa); we further proved this with later experiments.

### 3.2. SEM Analysis

The SEM images of ZnO thin film samples deposited at different sputtering powers are shown in Figure 4a–e. With an increase in power, the crystallized grains of the films tended to gradually get bigger. When the sputtering power was 100 W (Figure 4c), the surface of the ZnO film samples was flat and uniform; at 150 W (Figure 4d), the ZnO film samples had a large particle size and high growth rate; and at 200 W (Figure 4e), the surface of the ZnO film sample was rough and the grain was compressed. High power could improve the surface state density by depositing bigger grains. However, the surface structure became rough when the power was increased to 200 W. High-energy materials corrode the film surface and affect the surface morphology of the ZnO film sample. Therefore, at 100 W and 150 W, the film samples were flat and uniform. Similarly, it can be observed from the results in Figure 5a–e that when Ar:O_2_ = 15:10 (Figure 5c) and Ar:O_2_ = 25:10 (Figure 5d), the surfaces were relatively dense. The surface under the three sputtering pressures was also relatively dense, as shown in Figure 6a–c. Since the maximum sputtering pressure in our laboratory could only be adjusted to 1.0 Pa, and considering the best working state of the equipment, 0.7 Pa (Figure 6b) was more suitable. The SEM micrographs of ZnO films verified the rationality of the XRD results.

### 3.3. EDS Analysis

EDS analysis was carried out in order to further determine the elemental compositions of the obtained ZnO films. The EDS energy spectra of the ZnO films grown on GSS substrates at different sputtering powers are shown in Figure 7a–e. The quality score and atomic percentage of ZnO changed very little when the power was varied from 65 W to 200 W, and the measured value was close to the theoretical value. This showed that the deposited films were mainly composed of zinc and oxygen. In comparison, the mass fraction and atomic percentage were closer to the theoretical values when the power was 85 W or 100 W. Similarly, we can observe from Figure 8a–e that the composition was closer to the theoretical value at Ar:O_2_ = 5:10 and from Figure 9a–c that the optimum sputtering pressure was 0.7 Pa. Although these conclusions deviate from the XRD results, the XRD results are still reasonable considering the primary and secondary factors.

### 3.4. Analysis of Piezoelectric Properties 

In order to study the piezoelectric properties of the prepared ZnO films, the upper and lower electrodes were deposited under the same conditions; the structure is shown in Figure 1b. To ensure that the measurement accuracy was closer to the actual value, repeated measurements for each sample were carried out. Six measurements were made at different positions on the sample, and the mean value was calculated. The measurement results are shown in Table 4; the maximum standard deviation of all samples was ±0.03 pC/N.

In order to more intuitively show the influence of a single factor on *d*_33_, a relationship curve between the different parameters of each factor and the average value of *d*_33_ was drawn, as shown in Figure 10. Figure 10a shows the variation of the piezoelectric coefficient value *d*_33_ as a function of sputtering power.

The ZnO films deposited at a low sputtering power had a low *d*_33_ value of 2.62 pC/N. With the increase in sputtering power, the *d*_33_ coefficient value followed an increasing trend, and the maximum value reached 4.48 pC/N. Combined with Figure 3a,b, these results show that the piezoelectric properties are directly related to crystal orientation. When the power reached 200 W, the piezoelectric coefficient value decreased to 3.78 pC/N. This shows that 100 W and 150 W were relatively ideal sputtering powers in the studied parameter range. Similarly, Figure 10b shows that when the argon–oxygen ratio was increased from 5:10 to 30:10, the *d*_33_ coefficient value first increased and then decreased. When Ar:O_2_ = 10:10, Ar:O_2_ = 15:10, and Ar:O_2_ = 25:10, the *d*_33_ values were 4.78 pC/N, 4.85 pC/N, and 5.12 pC/N, respectively. Combined with Figure 3c,d, these results show that Ar:O_2_ = 25:10 was the best argon–oxygen ratio within the studied sputtering value range. Figure 10c shows that when the sputtering pressure was 0.7 Pa, the maximum *d*_33_ coefficient value was 4.57 pC/N. It can be seen from the above analysis that the *d*_33_ coefficient value of ZnO films fabricated under the current conditions is less than the data recorded in the literature [30,31]. Some reasons for this could be the accuracy of the sputtering equipment, different substrates, and different sputtering conditions. However, the current test results can meet the application requirements for the fabrication of smart bolts.

### 3.5. Film-to-Substrate Adhesion Analysis

Scratch experiments were carried out on samples deposited under different sputtering parameters. Figure 11 shows the scratch micrograph of a sample deposited under one of the parameters. The initial crack that appeared at load Fc1 is shown in Figure 11a, and Figure 11b shows the ZnO film completely peeled off at Fc2. The magnitude of adhesion strength between ZnO and GSS was determined by evaluating the critical load. 

Figure 12 shows the acoustic signal and adhesion strength curve of the scratch test on ZnO films deposited under different sputtering powers. The acoustic signal curves from the scratch test of the ZnO films are shown in Figure 12a–e. When the sputtering power was 65 W, 85 W, 100 W, 150 W, and 200 W, the adhesion between the ZnO film and GSS substrate was 3341, 3417, 3459, 3482, and 3521 mN, respectively. The variation curve of ZnO film adhesion with power is shown in Figure 12f. The adhesion of ZnO films increased gradually with the increase in sputtering power. An increase in sputtering power increases the deposition rate of the film, and the film thickness will increase under the same sputtering time [21]. In addition, as the sputtering power increases, the probability of the bombardment of target atoms and clusters increases, resulting in greater energy deposition on the surface of the substrate [32].

Figure 13 shows the acoustic signal and adhesion strength curve from the scratch test of ZnO films deposited under different argon–oxygen ratios. By analyzing the acoustic emission signal, it can be seen that when Ar:O_2_ was 5:10, 10:10, 15:10, 25:10, and 30:10, the adhesive strength of the film was about 4025, 4135, 4260, 4412, and 4220 mN, respectively. Through the comparison of ZnO films under different Ar:O_2_ ratios, it was demonstrated that with the increase in Ar:O_2_, the adhesion between the ZnO film and GSS substrate first increased and then decreased; when Ar:O_2_ = 25:10, the maximum adhesion force was 4412 mN. Combined with the XRD and SEM analysis results, it can be seen that when Ar:O_2_ = 15:10 or 25:10, the diffraction peak (002) of the film was strong and the crystallinity of ZnO film was improved; thus, the film displayed strong adhesion. However, under other conditions, the diffraction peak (002) is relatively weak, and the crystallinity of the film is poor; thus, the adhesion force is weak [33].

The acoustic signal and adhesion strength curve from the scratch test of ZnO films deposited under different sputtering pressures are shown in Figure 14. According to the scratch test data in Figure 14a–c, when the working pressure was 0.5, 0.7, and 0.9 Pa, the adhesion was 3985, 4251, and 4061 mN respectively. Figure 14d shows that the adhesion first increased and then decreased. When the sputtering pressure was increased to 0.9 Pa, the adhesion decreased. This may be due to the insufficient energy of the sputtering material to reach the substrate. When the sputtering pressure is high, frequent collisions between particles reduce the energy used for diffusion, resulting in poor adhesion between the ZnO and GSS substrate [34].

The scratch test results show that the ZnO/GSS structure has several favorable prosperities, such as high compressive strength, high fracture strength, and the absence of any breakage phenomenon.

## 4. Conclusions

Different series of ZnO films were fabricated using RF magnetron sputtering. From the analysis of XRD, SEM, EDS, piezoelectric properties, and film-to-substrate adhesion, it was concluded that microstructure is the critical factor on which the piezoelectric coefficient and adhesion of ZnO films depend. This microstructure can be optimized by selecting suitable values for the sputtering power, argon–oxygen ratio, and sputtering pressure. The optimum process for fabricating ZnO films by magnetron sputtering is as follows: a sputtering power of 150 W, Ar:O_2_ = 25:10, and a sputtering pressure of 0.7 Pa. In these conditions, the maximum d_33_ coefficient value was 5.12 pC/N, and the maximum value of film-to-substrate adhesion between the ZnO films and GSS substrate was 4220 mN. The XRD results showed that all film samples had a (002) crystal orientation at 2θ = 34.4°, which confirmed the development of a polycrystalline hexagonal-like structure. The results presented in this paper provide a theoretical basis for the application of ZnO piezoelectric films in the field of health monitoring with piezoelectric sensors.

## Figures and Tables

**Figure 1 micromachines-13-00639-f001:**
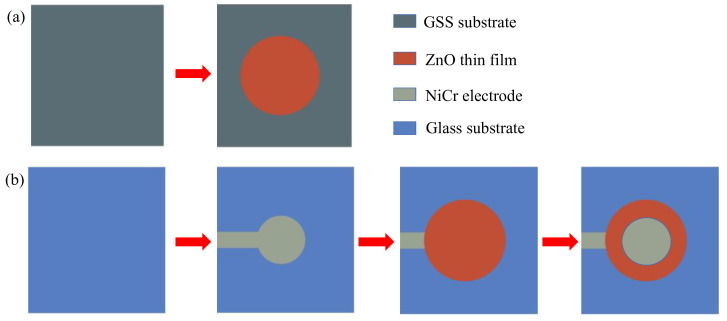
Schematic diagram of (**a**) the deposition process on GSS substrates and (**b**) the deposition process on glass substrates.

**Figure 2 micromachines-13-00639-f002:**
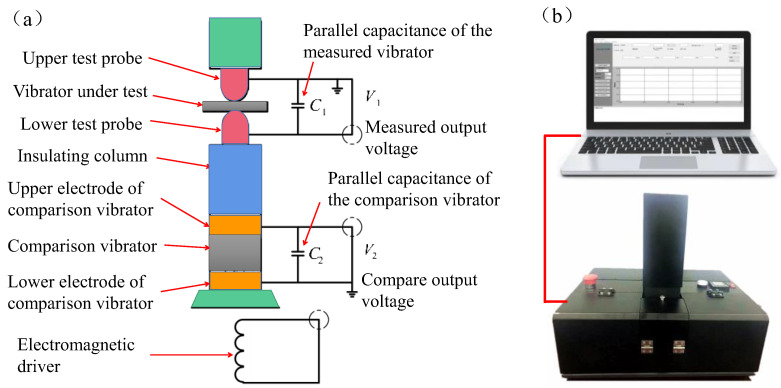
(**a**) Schematic diagram of the quasi-static test method. (**b**) Piezoelectric coefficient (*d*_33_) measuring instrument.

**Figure 3 micromachines-13-00639-f003:**
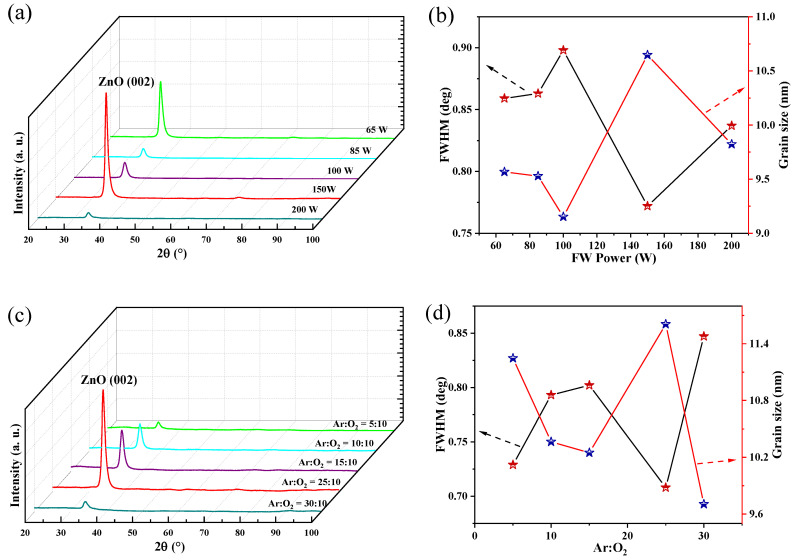
(**a**,**c**,**e**) XRD spectrum of ZnO films and (**b**,**d**,**f**) the FWHM and grain size of ZnO thin films at the (002) peak in different deposition conditions.

**Figure 4 micromachines-13-00639-f004:**
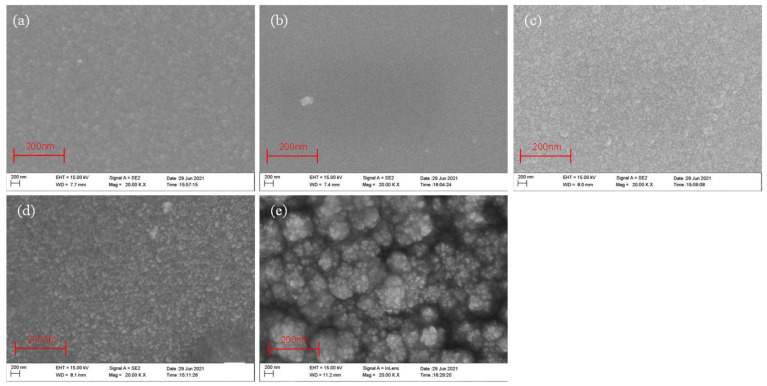
SEM micrographs of ZnO films on GSS substrate at the different sputtering powers: (**a**) PW65, (**b**) PW85, (**c**) PW100, (**d**) PW150, and (**e**) PW200.

**Figure 5 micromachines-13-00639-f005:**
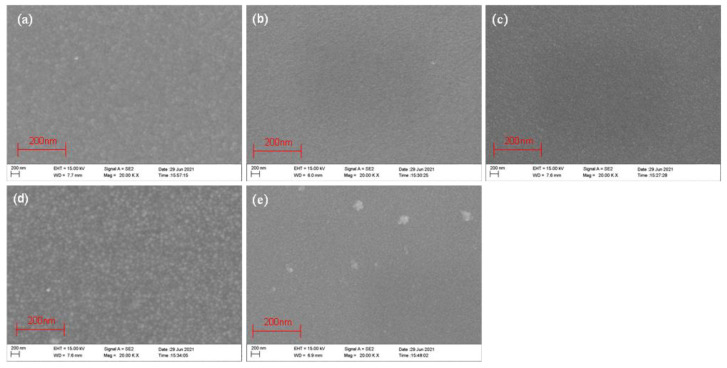
SEM micrographs of ZnO films on GSS substrate at the different argon–oxygen ratios: (**a**) R5-10, (**b**) R10-10, (**c**) R15-10, (**d**) R25-10, and (**e**) R30-10.

**Figure 6 micromachines-13-00639-f006:**

SEM micrographs of ZnO films on GSS substrate at the different sputtering pressures: (**a**) PA05, (**b**) PA07, and (**c**) PA09.

**Figure 7 micromachines-13-00639-f007:**
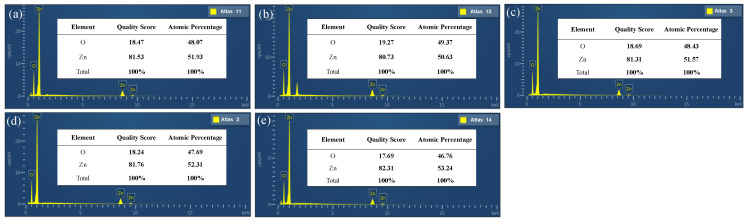
Typical EDS spectra of the reference samples at different sputtering powers: (**a**) PW65, (**b**) PW85, (**c**) PW100, (**d**) PW150, and (**e**) PW200.

**Figure 8 micromachines-13-00639-f008:**
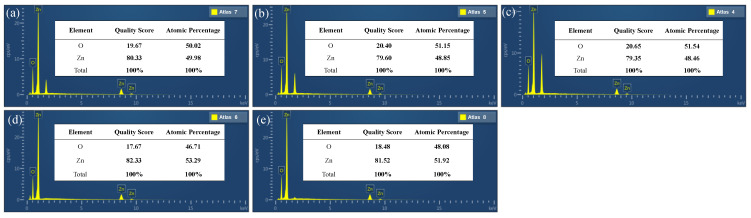
Typical EDS spectra of the reference samples at different argon–oxygen ratios: (**a**) R5-10, (**b**) R10-10, (**c**) R15-10, (**d**) R25-10, and (**e**) R30-10.

**Figure 9 micromachines-13-00639-f009:**
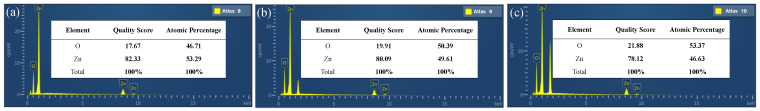
Typical EDS spectra of the reference samples at different sputtering pressures: (**a**) PA05, (**b**) PA07, and (**c**) PA09.

**Figure 10 micromachines-13-00639-f010:**
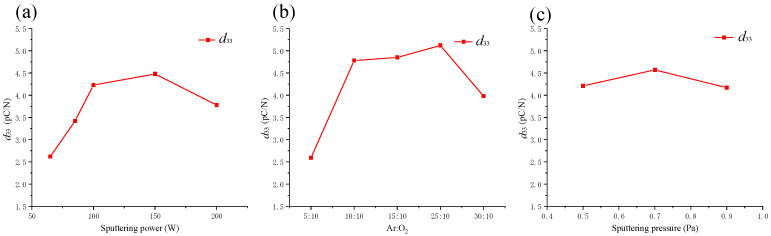
Relationship between each factor and the average value of the piezoelectric coefficient. (**a**) *d*_33_ of ZnO films under different sputtering powers. (**b**) *d*_33_ of ZnO films under different argon–oxygen ratios. (**c**) *d*_33_ of ZnO films under different sputtering pressures.

**Figure 11 micromachines-13-00639-f011:**
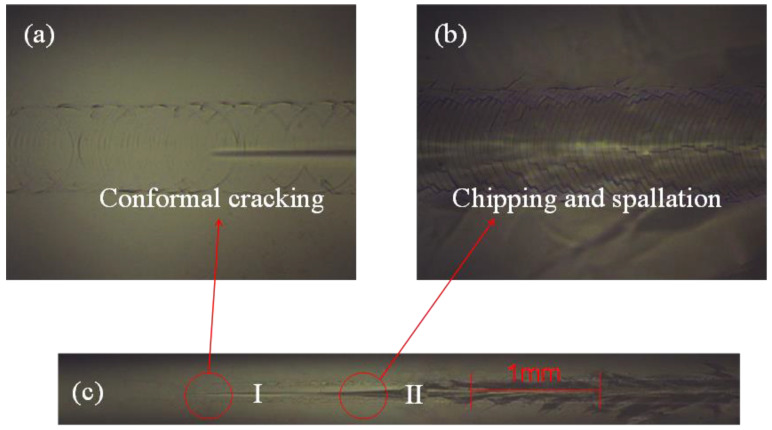
Examples of scratching on the ZnO thin films deposited on the GSS substrate (Power: 100 W; Ar:O_2_ = 25:10; Pressure: 0.7 Pa), (**a**) the magnified views of conformal cracking, (**b**) the magnified views of chipping and spallation, (**c**) optical images of scratch tracks.

**Figure 12 micromachines-13-00639-f012:**
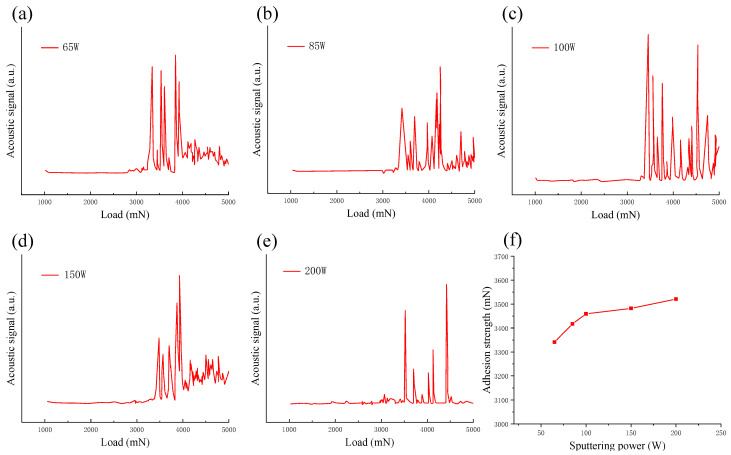
(**a**–**e**) Acoustic signal and (**f**) adhesion strength curve from the scratch test of ZnO film under different sputtering powers.

**Figure 13 micromachines-13-00639-f013:**
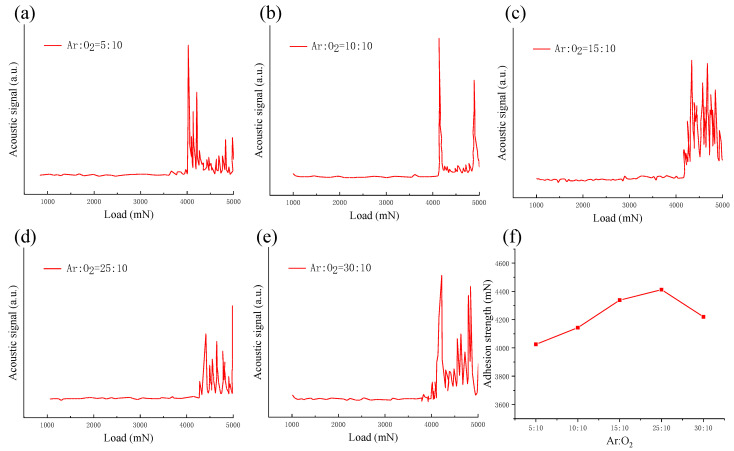
(**a**–**e**) Acoustic signal and (**f**) adhesion strength curve from the scratch test of ZnO film under different argon–oxygen ratios.

**Figure 14 micromachines-13-00639-f014:**
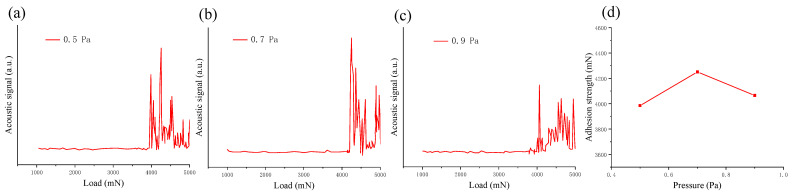
(**a**–**c**) Acoustic signal and (**d**) adhesion strength curve from the scratch test of ZnO film under different sputtering pressures.

**Table 1 micromachines-13-00639-t001:** Experimental materials and specifications.

Target	Target Size	Sputtering Gas	Substrate	Target-to-Substrate Distance	Sample Size
Zn 99.999%	φ100 × 3 mm	Ar and O_2_	GSS and glass	65 mm	18 × 18 × 0.6 mm

**Table 2 micromachines-13-00639-t002:** Properties of the GH4169 superalloy steel substrate.

Density (g/m^3^)	Young’s Modulus (GPa)	Thickness (mm)	Coefficient of Thermal Expansion (ppm/°C)	Melting Point (°C)
8.2	199.9	0.5	11.8 (20–100 °C)	1260–1340

**Table 3 micromachines-13-00639-t003:** Experimental details of ZnO films fabricated using different process parameters.

Sample Series	Sample Identity	Power (W)	Argon–Oxygen Ratio	Working Pressure (Pa)	Ambient Environment/Pressure (pa)
Series-I	PW65	65	15:10	0.7	1.0 × 10^−3^
	PW85	85	15:10	0.7	1.0 × 10^−3^
	PW100	100	15:10	0.7	1.0 × 10^−3^
	PW150	150	15:10	0.7	1.0 × 10^−3^
	PW200	200	5:10	0.7	1.0 × 10^−3^
Series-II	R5-10	100	5:10	0.7	1.0 × 10^−3^
	R10-10	100	10:10	0.7	1.0 × 10^−3^
	R15-10	100	15:10	0.7	1.0 × 10^−3^
	R25-10	100	25:10	0.7	1.0 × 10^−3^
	R30-10	100	30:10	0.7	1.0 × 10^−3^
Series-III	PA05	100	15:10	0.5	1.0 × 10^−3^
	PA07	100	15:10	0.7	1.0 × 10^−3^
	PA09	100	15:10	0.9	1.0 × 10^−3^

**Table 4 micromachines-13-00639-t004:** Measurement results for the piezoelectric coefficient at different process parameters.

Sample Name	Average Value of Piezoelectric Coefficient							
	Test 1	Test 2	Test 3	Test 4	Test 5	Test 6	Average	STD
PW65	2.71	2.49	2.85	2.42	2.58	2.65	2.62	±0.07
PW85	3.39	3.54	3.41	3.71	2.98	3.46	3.42	±0.11
PW100	4.44	4.61	3.89	4.01	4.46	3.97	4.23	±0.14
PW150	4.88	3.98	4.47	4.55	4.45	5.12	4.48	±0.18
PW200	3.87	3.45	4.15	3.47	4.08	3.68	3.78	±0.14
R5-10	2.68	2.43	2.35	2.55	2.76	2.76	2.59	±0.08
R10-10	4.93	4.75	4.64	5.16	4.96	4.21	4.78	±0.15
R15-10	4.47	5.49	5.39	4.42	4.88	4.47	4.85	±0.22
R25-10	4.96	5.39	5.48	4.88	5.28	4.71	5.12	±0.14
R30-10	3.85	4.11	3.58	4.29	4.39	3.69	3.98	±0.15
PA05	4.31	3.98	4.18	4.39	3.97	4.41	4.21	±0.09
PA07	4.45	4.74	4.39	4.86	4.57	4.42	4.57	±0.09
PA09	3.95	3.89	4.16	4.41	4.35	4.28	4.17	±0.10

## Data Availability

Some or all data, models, or code generated or used during the study are available from the corresponding author by request.

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
