# Peer review of "Influence of ZnO Film Deposition Parameters on Piezoelectric Properties and Film-to-Substrate Adhesion on a GH4169 Superalloy Steel Substrate"

_micromachines, 2022, doi:10.3390/mi13040639_

Round 1
Reviewer 1 Report
All the comments and suggestions are addressed, and I recommend the manuscript to be accepted in present form.
Author Response
Dear Professor,
Thank you very much for your valuable comments on the paper. Through revision, the article has reached a higher level and improved my writing level. Thank you!
Reviewer 2 Report
The authors improved well the manuscript after corrections. The manuscript can be considered suitable for publication after considering the following remarks:
- Adding of a scale bar (in either in mm or microns) on figures 11 (page 11) of scratch testing
- page 2, line 130 : D3 instead of D33
Author Response
Dear Professor,
Thank you very much for your valuable comments. I have revised these two contents. Thank you.
Best wishes,
Guowei Mo
This manuscript is a resubmission of an earlier submission. The following is a list of the peer review reports and author responses from that submission.
Round 1
Reviewer 1 Report
In the manuscript titled as “Influence of ZnO films deposition parameters on piezoelectric properties and film-to-substrate adhesion on GH4169 superalloy steel substrate‘’, ZnO thin films were grown on GSS substrate by magnetron sputtering, and the effects of sputtering power, argon oxygen ratio, and sputtering pressure on piezoelectric coefficient and the adhesion of sputtered film were studied. However, this manuscript lacks innovation and logical integrity, and also does not meet the requirements of Micromachines, so I suggest rejecting the manuscript and offer the following comments for consideration.
1. The quality of the all pictures in the manuscript are not good enough.
2. In the figure 3 (a)(c)(e), the grid of X-Y plane should be added.
3. In the section of introduction, the innovation of this research needs to be further clarified, especilly the using of GSS.
4. The detailed information of GH4169 superalloy steel should be provided.
5. Add the performance comparison of the sample with previous references.
6. Moderate English changes required.
Reviewer 2 Report
In the article of G. Mo et al. entitled “Influence of ZnO films deposition parameters on piezoelectric properties and film-to-substrate adhesion on GH4169 superalloy steel substrate”, the authors addressed an experimental work to process ZnO thin film by RF magnetron sputtering on GH4169 superalloy steel (GSS) and glass substrates with NiCr bottom electrodes. The authors mention a perspective of their work as a premise of highly piezoelectric thin film to be applied to health monitoring sensors. The authors applied different methods of characterization of the obtained thin film: to estimate the piezoelectric response, to indirectly evaluate the adhesion of the films by scratch testing and cracks signatures under load. As general conclusion of the analysis detailed below, the article has serious flaws, additional experiments are needed. The article is not all suited for publication, and it has to reconsider for the whole before to be resubmitted to a journal.
State-of-the-art part:
The prior art mentioned numerous use of magnetron sputtering to process ZnO thin films, with also perspectives (and final devices processed) addressing the use of these films as strain gauges for SHM (Structure Health Monitoring) by piezoresitive, piezoelectric and/or piezotronic effects. So, in this context the study proposed there by G. Mo et al. did not reveal novelties in terms of methodologies and experimental results for thin film ZnO with piezoelectric response. Additionally, although the authors envision their ZnO thin film in health monitoring sensors, they did not explain in which configuration they will use the film to get a sensor. One cannot estimate a clear novelty and scientific impact of the work.
Additionally, here is a list of points to consider by the authors:
- ZnO has not steady chemical properties as mentioned by the authors, page 1 line 32. ZnO is well known to be sensitive to corrosion by moisture in the atmosphere. For long term use of ZnO in sensors, a passivation layer is mandatory. Else one observed a drift over time of the electrical performances such as the conductivity and piezoelectric coefficient value.
- Method of fabrication of ZnO by ALD (Atomic Layer Deposition) is missing in the state-of-the-art, page 1 lines 37-39. ALD is indeed a very used manufacturing method of deposition for ZnO thin films (in labs and also in industry) with high conformality and high level of adhesion on several kind of substrates’ materials. The authors should consider this way of ZnO thin film ALD deposition with some related references.
- “Compatibility” term must be detailed in the sentence “It is very critical to study the compatibility of ZnO films besides the effects of deposition time and deposition temperature”, page 2 lines 47-48
- More quantitative value is expected for the meaning of “higher piezoelectric properties”, page 2 line 58
- What is the meaning of “diameter” in the sentence “Cimpoiasu et al. deposited ZnO films on diameter silicon wafers substrate (…)” ? , page 2 line 65
- The “etc” term used many times in the text of the article is not very convenient and should be avoided considering that the authors have to considered the most relevant cases for the understanding in the article, page 2 line 69, page 3 line 104, page 8 line 227.
- The term “bonding” is not well appropriate for a thin film deposited on a substrate. Adhesion term has to be employed instead. Page 2 line 69
- The term “anxious” is not well appropriate and must be replaced by a more relevant wording. Page 2 line 72
- The sentence “Few public reports of the search result for sputtering piezoelectric ZnO films on GSS substrates” has to be reconsidered by the authors for grammar (no verb), and also about the references to cite about the “Few public reports”.
Experimentals:
In the experimental part, a certain number of details are missing to assess the methodologies and the results of the characterizations.
- Adhesion performances represent a major concern of this study. Scratch testing is not enough. Testing for adhesion must be aligned on a standard to be relevant. This is especially important to consider this standard for adhesion testing that the authors consider their film development for future health monitoring sensors as they mentioned in page 3 line 106. The tape test following the ASTM D3359 standard (“Standard Test Methods for Measuring Adhesion by Tape Test”) must be considered by the authors to evaluate the adhesion between the ZnO film and the substrate.
- Piezoelectric response of the ZnO thin film represents another major concern of this study. Wurtzite ZnO has indeed anisotropic piezoelectric responses, but it is also well known to have semiconducting properties with the very difficult challenge to manage the leakage current during piezoelectric characterization. Could the authors detail more their methodology of characterization, and not only the equipment used (page 4 lines 113-116), of the ZnO thin film to get the piezoelectric coefficients (listed on table 3, page 8) without artefacts of measurement due to this leakage current? Related electrical equation to quantify the d33 coefficient would be also appreciated for understanding of the methodology used.
- The use of NiCr as bottom electrode on glass substrate needs to be explain by the author. Role of the Chromium on the adhesion on glass ?
- The vision of the authors to apply their film development for future health monitoring sensors as they mentioned (page 3 line 106) point the question about the scalability of their process to process the ZnO thin film on large parts of structures with 3D shaping (e.g. parts of aircraft such doors, wings). In this context of structure health monitoring, could the authors propose some perspective of scalability of their process with also the limitations that can occur in term of quality as the homogeneity of deposition and electrical performance ?
- The authors have to be more quantitative when they discussed about the relationship between the sputtering power and deposition rate and thickness of the film (page 9, lines 248 250). Could they add some values there such as a graph of the deposition rate and thickness against the sputtering power?
- For figures 4(a)-(e), the different values of the sputtering power used for each case is not explicitly referenced (both in the caption of the figure and also in the text page 5 lines 163-165)
- For figures 5(a)-(e), the different values of the argon:oxygen used for each case is not explicitly referenced (both in the caption of the figure and also in the text page 6 lines 170-171)
- For figures 6(a)-(c), the different values of the sputtering pressure used for each case is not explicitly referenced (both in the caption of the figure and also in the text page 6 lines 172-175)
- The quality of the figures 7 to 9 of the EDS spectra is execrable and not at all suited to be published. The authors have to optimize the readability of these figures (e.g. X scale in energy, labelling of the atom )
- Adding of the scale bar on figures 11 (page 9) of scratch testing would be appreciated.
English language issues:
The text contains a lot of English typos, misplaced verb tenses (past, present), sentences without main verbs and awkward expressions. The text needs extensive editing for English, and should be proofread by a native English speaker.
Typo errors:
- “Scherrer” instead of “Scheler” , page 4 line 142
- “Results” instead of “rusults” page 8 line 208